# PeerJ

# A localized PCR inhibitor in a porcelain crab suggests a protective role

Mahmoud A. El-Maklizi[1], Amged Ouf[1,2], Ari Ferreira[2], Shahyn Hedar[1] and Edwin Cruz-Rivera[3]

[1] Biology Department, The American University in Cairo, New Cairo, Egypt
[2] Biotechnology Program, The American University in Cairo, New Cairo, Egypt
[3] Biological Sciences Program, Asian University for Women, Chittagong, Bangladesh

## ABSTRACT

A number of polymerase chain reaction (PCR) inhibitors have been identified from biological and environmental samples. By and large, such substances are treated as random nuisances and contaminants with alternate functions; their inhibitory effects on DNA replication being a coincidental property of their molecular structure. Here, we demonstrate the presence of a localized PCR inhibitor in the foregut of the porcelain crab *Petrolisthes rufescens* (Anomura: Porcellanidae) from the Red Sea. The inhibitor precluded amplification of 28s, 16s and 18s gene sequences effectively but lost activity at $10^{-2}$ dilutions from initial concentration. Heat treatment was ineffective in arresting inhibition and spectrophotometric techniques suggested that the inhibitor was not a melanin-type compound. The compound was not detected from midgut, hindgut, or gills of the crab. Activity of the inhibitor was precluded when samples were treated with suspensions from the midgut, suggesting that enzymatic degradation of the inhibitor likely happens at that part of the gut. As many microbial pathogens invade their hosts via ingestion, we suggest the presence of the localized inhibitor could carry a defensive or immunological role for *P. rufescens*. The identity of the inhibitory molecule remains unknown.

## INTRODUCTION

Polymerase chain reaction (PCR) is a powerful, rapid method for the diagnosis of microbial infections and genetic diseases, the detection of microorganisms in environmental and food samples, forensics, and the amplification of DNA sequences for phylogenetic and ecological studies (*McCartney, 2002*; *Rådström et al., 2004*; *Maurer, 2011*; *Alaeddini, 2012*). Application of this tool to environmental and biological samples is often hampered by the presence of unknown inhibitors that block one or more of the steps yielding DNA amplification, and a variety of organic and inorganic inhibitors have been detected or isolated (*Wilson, 1997*; *Rådström et al., 2004*; *Schneider, Enkerli & Widmer, 2009*; *Maurer, 2011*; *Alaeddini, 2012*; *Schrader et al., 2012*). Inhibitors can affect any step of PCR and normally act by reducing or arresting cell lysis required to extract DNA, by degrading nucleic acids, by binding to, and blocking capture of, nucleic acids, or by inhibiting the action of polymerases in amplifying target DNA (*Wilson, 1997*; *Alaeddini, 2012*;

Corresponding author
Edwin Cruz-Rivera,
edwin.rivera@auw.edu.bd

*Schrader et al., 2012*). The different mechanisms are a reflection of the diversity of chemical structures of inhibitors, many of which are widespread in nature, including phenolic compounds and humic acids, carbohydrates like glycogen, fats, and various proteins (*Wilson, 1997*; *Rådström et al., 2004*; *Maurer, 2011*; *Alaeddini, 2012*; *Schrader et al., 2012*). In the case of studies with animals, skin, muscle and blood components, including pigments (e.g., hemoglobin, melanin) are known to block PCR (*Yoshii et al., 1993*; *Akane et al., 1994*; *Belec et al., 1998*; *Eckhart et al., 2000*). For such cases, sample preparation often requires dilution of the samples containing the inhibitor, pretreatment of extracted DNA aliquots with columns of specific molecular affinities to bind inhibitors, immunocapture of cells, the use of two-phase aqueous systems to separate cells or DNA from inhibitors, the addition of substances that precipitate the suspected inhibitor, or the alternative use of different polymerases with varying sensitivities to inhibition (*Rådström et al., 2004*; *Schneider, Enkerli & Widmer, 2009*; *Maurer, 2011*; *Alaeddini, 2012*; *Schrader et al., 2012*).

In light of the ubiquity of inhibitors and the need for correcting steps in processing, the vast majority of molecular studies finding PCR inhibitors have largely considered such substances as coincidental contaminants in the samples (see above reviews). Given that many of the inhibitors isolated to date have other known biological functions, interference of PCR by these substances is considered incidental. This has resulted in an overall lack of work aimed at understanding the potential role of such inhibitors in controlling the replication of DNA foreign to the organisms producing them. Yet, it is plausible that molecules blocking DNA replication may have other functions, for example, as cytostatic or cytotoxic agents arresting cell division of microbial pathogens.

In this study, we report the presence of a yet-unknown inhibitory molecule that is localized in the foregut of the porcelain crab *Petrolisthes rufescens* (Anomura: Porcellanidae). This species is a filter feeder commonly found in the intertidal zones of the Red Sea, Persian Gulf, East Africa, Arabian Sea, and Indian Ocean (*Ahmed & Mustaquim, 1974*; *Haig, 1983*; *Siddiqui & Kazmi, 2003*; *Werding & Hiller, 2007*). Preliminary research (Fig. S3) demonstrated a rich microbial flora in other parts of this animal. However, efforts to isolate and amplify microbial DNA from the foregut consistently failed. This led to the hypothesis that a compartmentalized inhibitor of DNA replication, which could play a protective role against ingested microbes, was present in the foregut of the crab. To that effect we quantified the inhibitory activity of foregut extracts against eukaryotic and prokaryotic DNA amplification, the lower limits of activity for the inhibitor, the degradation of the inhibitor within the crab digestive tract, and preliminarily assessed the identity of the inhibiting molecule.

## MATERIALS AND METHODS

*Petrolisthes rufescens* (Heller, 1861) were collected at low tide from underneath rocks in the Ain Sukhna intertidal, Gulf of Suez, Egypt (29°57′N 32°32′E). Individuals were kept alive in plastic tubs with fresh seawater and transported to The American University in Cairo, where they were individually placed in plastic bags and frozen at −20 °C. Animals used were mature adults. Despite its broad geographic distribution and potentially dense

**Table 1  Yields of DNA from crab parts.** DNA extracted (mean ± 1SE) from different dissected parts of the porcelain crab *Petrolisthes rufescens* based on NanoDrop readings ($N = 6$).

| Body part | DNA concentration (ng/µl) |
|---|---|
| Foregut | $0.769 \pm 0.478$ |
| Midgut | $1.919 \pm 1.541$ |
| Hindgut | $0.487 \pm 0.140$ |
| Gills | $1.073 \pm 0.704$ |

populations (e.g., in our collection sites, Fig. S1), very little is known about the basic biology and ecology of this species (*Ahmed & Mustaquim, 1974*; *Yaqoob, 1974*; *Paul, Sankolli & Shenoy, 1993*), but it is a filter feeder that traps floating particles by extending its plumose third maxillipeds like other porcelain crabs (*Achituv & Pedrotti, 1999*; *Valdivia & Stotz, 2006*; *Riisgård & Larsen, 2010*). We have maintained specimens alive in recirculated seawater by feeding them on a mixture of live *Artemia salina* nauplii and finely-ground fish food flakes for over seven months.

Extraction of DNA was performed on frozen crabs ($N = 6$ for our preliminary extractions to quantify DNA yield (Table 1) and $N = 3$ for all experiments after). These were dissected to separate foregut, midgut, hindgut, muscle, and gills under a dissecting microscope, as necessary (see below). During dissection, animals were placed on a Petri dish kept cold above a layer of ice. DNA from each tissue was extracted using the DNEasy tissue extraction kit (Qiagen cat # 69504) and the DNEasy spin column protocol. Each tissue sample was placed in a 1.5 ml microcentrifuge tube and ground under 180 µl of Buffer ATL, before adding 40 µl of proteinase K and incubating at 56 °C for 1 h. Proteinase K was added at twice the specified concentration (unless otherwise indicated below) because we expected the digestive system environment to be high in proteins that could potentially inhibit PCR. Samples were periodically vortexed during digestion, and then finally for 15 min, before adding 200 µl of Buffer AL and 200 µl of ethanol, and vortexing again. DNA was purified by centrifuging serially in a DNeasy Mini spin column in three one-minute steps, including transfer to two purification buffers, as per manufacturer specifications. All transfer procedures were performed inside a sterile hood. Separate amplifications were periodically performed (without crab or fish tissues) to assess contamination of the buffers and sterile water used in the procedures.

In addition to the crab organs, fish muscle DNA (from the goatfish *Upeneus nigromarginatus*) was extracted using the same protocols and served as positive control in various experiments as explained below. Henceforth, the use of the word "extract" will refer to aliquots resulting from DNA extraction procedures. A NanoDrop 3300 fluorospectrometer (Thermo Scientific, Waltham, Massachusetts, USA) was used to quantify DNA extracted using the Quant-iT PicoGreen dsDNA assay kit (Life Technologies cat # P11496; Thermo Fisher, Waltham, Massachusetts, USA). As per manufacturer specifications, this instrument and technique can detect DNA concentrations down to 0.001 ng/µl. To assess the efficiency of our DNA extraction protocols, preliminary quantification was performed on parts from 6 randomly selected crabs and on the fish DNA used as control. Two serial

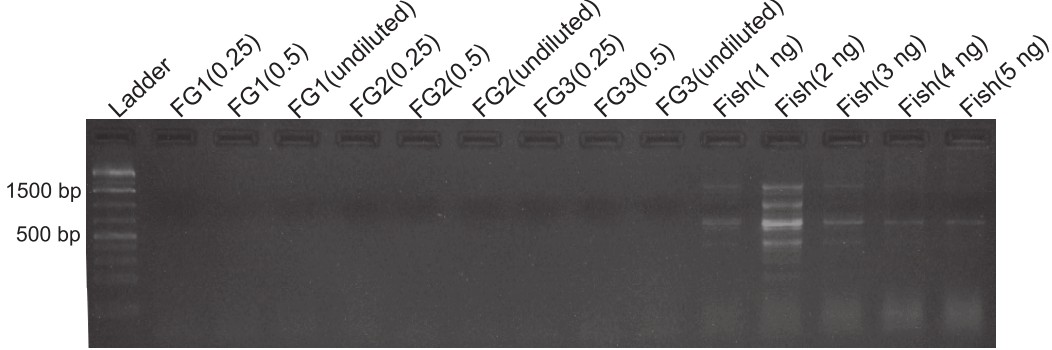

**Figure 1 DNA amplification at different concentrations in foreguts and fish extracts.** Amplification products of 28s primers from different *Petrolisthes rufescens* foreguts (FG) and fish muscle (see Materials and Methods). Extracted DNA aliquots were added to reactions at different amounts in order to assess potential inhibition due to DNA template concentrations. Undiluted foregut DNA masses in PCR reactions were 18.7 ng (FG1), 4.1 ng (FG2), and 0.93 ng (FG3). From these, 50% (0.5) and 25% (0.25) dilutions were also tested. Fish DNA controls contained 1–5 ng per PCR reaction.

dilutions per sample were done, $10^{-1}$ and $10^{-2}$, and an equal volume of the dye was added to each dilution before measuring absorbance 530. We also used NanoDrop fluorospectrometry throughout the study to standardize the amount of DNA in our PCR reactions to 2 ng of extracted crab DNA and 2 ng of control fish DNA. Early experiments (e.g., Fig. 1) showed the best amplification of fish DNA in reactions using this amount (see below). This is also consistent with other studies using 2 ng of DNA per 20–25 µl PCR reaction, although amplification can be observed at much lower concentrations (*Andrade et al., 2012*; *Jin et al., 2012*; *Bernal-Martínez et al., 2013*; *Batmalle et al., 2014*; *Okeke et al., 2014*).

To assess the presence and distribution of the inhibitor various PCR-based experiments were designed. All PCR reactions were developed on 1.0–1.7% agarose gels stained with ethidium bromide and the intensity of the bands obtained was observed under UV light in an ImageQuant 300 (General Electric Healthcare, Little Chalfont, United Kingdom) gel imaging system. PCR conditions for each primer used are provided in Table S1. In the first experiment, we assessed the sensitivity of our protocols and the possibility that the previously observed lack of DNA amplification in the foregut could be due to either too high or too low concentrations of DNA, both of which can lead to false negatives in PCR (*Wilson, 1997*; *Alaeddini, 2012*). Foregut extracts from individual crabs ($N = 3$) were diluted serially by decreasing the amount of extracted DNA added to the PCR reaction in a total volume of 20 µl per reaction. In this initial exploration, NanoDrop readings showed extracted DNA amounts of 18.7, 4.1, and 0.93 ng in undiluted crab foreguts (FG1, FG2, and FG3, respectively, Fig. 1). These amounts were twice serially diluted in half for each crab and the three concentrations were then used in PCR reactions. Thus this experiment tested amplification using foregut concentrations of ca. 19 (FG1) to 0.25 ng (0.25 of undiluted FG3) per reaction. As controls, five dilutions of fish DNA (5, 4, 3, 2 and 1ng per PCR reaction) were simultaneously run. Reactions were amplified using the 28s primers OI (5′-GTCTTTGCGAAGAAGAACA-3′) and DIB (5′-AGCGGAGGAAAAGAAACTAAC-3′) described in *Morrison et al. (2002)*.

We directly assessed the inhibitory activity of *P. rufescens* foreguts ($N = 3$) using a fixed amount of foregut DNA extract (2 ng) and control fish DNA (2 ng) added to the reactions. Using the same primers for amplification described above (28s), PCR reactions were run along a control containing 2 ng of fish DNA in the reaction, but no extracted crab foregut (positive control) and a negative control lacking any DNA. Prior assays showed positive amplification at even higher amounts of DNA than 4 ng (Fig. 1). Thus, failure to obtain amplified fish 28s DNA in the presence of crab foregut extracts would indicate inhibition or disruption of PCR compared to a successfully amplified 28s band in the control containing only fish DNA. For this experiment, crab extracts were diluted twice (as in the previous experiment), resulting in reactions with 2, 1 and 0.5 ng of foregut DNA.

To assess whether the inhibitor was systemic in the crabs, amplification was further performed on DNA extracted from various parts of the crabs (foregut, midgut, hindgut, and gills). For this experiment, we used universal bacterial primers and, therefore, the experiment also provided indirect information on the presence of microbes in the gut and other organs of *P. rufescens*. Thus the purpose of the experiment was twofold. First, it determined if inhibition was localized in the foregut of the crab. Second, using bacterial primers also assessed if inhibition was related to PCR specifically and not to difficulties amplifying nuclear genes such as 28s. The universal primers (*Fierer et al., 2007*) Bac8f (5′-AGAGTTTGATCCTGGCTCAG-3′) and Univ529r (5′-ACCGCGGCKGCTGGC-3′) were used on samples ($N = 3$) of: (1) midguts, hindguts, and gills pooled per crab and (2) foreguts, midguts, hindguts and gills pooled per crab. Samples were adjusted so that a total of 2 ng of DNA (based on NanoDrop readings) were used in each PCR reaction: 1 ng of DNA from the foregut mixed with a total of 1 ng of DNA from midgut, hindgut and gills together. A negative control (all PCR components except crab extract) was run simultaneously to assess potential microbial contamination of reagents during the experimental process.

The limits of activity of the inhibitor were assessed by progressively reducing the amount of foregut extract in PCR reactions containing 2 ng aliquots of fish DNA. For this, the amount of crab foregut extract was sequentially diluted in multiples of 10 from initial (2 ng) concentrations ($10^{-1}$, $10^{-2}$, $10^{-3}$, and $10^{-4}$) and then added to the PCR reactions, which were then amplified using the 18s primers 18E-F (5′-CTGGTTGATCCTGCCAGT-3′) and 18sR3 (5′-TAATGATCCTTCCGCAGGTT-3′) (*Kim & Abele, 1990*). Thus, fish DNA was amplified in the presence of undiluted foregut ($N = 3$) aliquots, plus four serial dilutions of these same aliquots, along with two positive controls (2 ng of fish DNA added alone) and a negative control (all PCR components except DNA from crabs or fish) for assessing contamination (using yet a different set of primers to assess generality of the inhibitor).

If the inhibitor served a functional role in the foregut, and its activity was generalized for DNA replication, it was expected that regulation of some sort would keep the compound from affecting other parts of the digestive tract, where it could inhibit other potentially beneficial microbes or cells. To test for this, two experiments were performed by comparing the inhibition of PCR by foregut extracts versus that of foreguts mixed with midgut

suspensions. Our hypothesis was that enzymatic degradation (or any other type of neutralization) of the inhibitor occurred in the midgut, the contiguous digestive chamber of the crab. Because our interest was to account for enzyme activity, we did not add proteinase K to the samples. Other protocols were kept the same. In the first experiment, undiluted extracts of foreguts (2 ng) were mixed with 2 ng control fish DNA. These were compared to samples from the same crabs in which foreguts and midguts were mixed in equal amounts before adding the fish DNA. All reactions were processed by using the 18s primers previously described. Foreguts and midguts were homogenized in 100 µl of Ultrapure, DNase and RNase free water (Invitrogen inc.). The foreguts were extracted as above.

The follow-up experiment aimed at establishing the minimum activity of the midgut suspensions that would arrest the activity of the inhibitor. This experiment followed the protocols above, but after the extraction, 50 µl of the foreguts were divided into five equal portions (10 µl each). To each portion an equal volume of either the midgut initial suspension from that same crab or one of its dilutions ($10^{-1}$, $10^{-2}$, $10^{-3}$, $10^{-4}$) was added. The mixtures were incubated at 42 °C for two hours on a shaking thermoblock. This temperature was chosen to represent a potential extreme observed in areas of Egypt during summertime. Crabs often spend several hours above water when the tide recedes, where they are surrounded by tan to dark rocks and gravel (Fig. S1). After the incubation period, another round of extraction was done on the mixtures. The extracted mixtures were then used in the 18s PCR. Each PCR reaction contained 2 µl of the mixture plus 3 µl of fish DNA. In total, each mixture of foregut and midgut from the same crab ($N = 3$) had five PCR reactions (15 PCR reactions in total). For this one and the previous assay samples were normalized using masses because the midguts were not extracted, but rather macerated in water and suspended in buffer, keeping us from normalizing using NanoDrop readings.

Because the identity of the inhibitor was unknown, we assessed qualitatively whether the molecule was a protein with a secondary, tertiary or quaternary structure. Three crab foreguts were individually heated in AE buffer using a thermoblock at 99 °C for one hour. The boiled foreguts were then extracted using the explained procedures and 2 ng of each product was added to 2 ng of fish DNA. PCR was then performed using 18s primers as previously done. By comparing simultaneously the three mixtures of pre-heated foreguts with fish DNA against a sample of the same fish DNA alone, we determined if heating degraded the inhibitory molecule, as would be expected from a complex protein (but see *Abu Al-Soud, Jönsson & Rådström, 2000*).

The possibility that the inhibitor was a melanin-type of pigment was also tested. Melanins occur in crustaceans and other invertebrates (*Söderhäll, 1982*; *Bandaranayake, 2006*; *Vázquez et al., 2009*; *Dubey & Roulin, 2014*), and are known to interfere with PCR (*Alaeddini, 2012*; *Schrader et al., 2012*). We assessed the possibility that the inhibitor was a melanin-related compound using the spectrophotometric approach developed by *Dörrie et al. (2006)*. Six treatment groups: crab foregut, midgut, hindgut, gills, and muscle, and fish muscle ($N = 4$) were extracted with the same protocols previously described and tested. A dilution of $20^{-1}$ for each sample using TE buffer was made in a total volume of 400 µl. The absorbance of each sample at 320 nm was measured using a UV spectrophotometer.

 

The measurement was repeated twice for each sample, using 196 µl each time and the two readings were subsequently averaged. The absorbances of the six groups were compared using one-way ANOVA. While this method does not quantify absolute melanin content in the absence of a melanin standard, or non-soluble melanin present in the tissues, it does provide reliable information on relative amount of melanin dissolved among samples (*Dörrie et al., 2006*; *Sánchez-Rodríguez et al., 2008*). A high absorbance of the foreguts in relation to other tissues would suggest the presence of either naturally-occurring melanins or a similar compound leached from tissues nearby the mouth parts.

## RESULTS

Although variation in extracted yield was high, all samples showed non-zero readings for DNA (Table 1). The concentrations ranged from a minimum of 0.003 (in one midgut) ng/µl to a maximum of 9.53 ng/µl (also in a midgut). Extracted DNA from the foreguts ranged from a minimum of 0.156 ng/µl to a maximum of 3.123 ng/µl. For comparison, the fish muscle sample used as control in the first dilution experiment yielded 1.141 ng/µl. On average the lowest amounts of DNA were obtained from the hindgut and foregut (Table 1).

Despite yielding comparable amounts of DNA to control fish DNA concentrations, foreguts did not amplify when 28s primers were used, regardless of the amount of aliquot added to the PCR reactions (Fig. 1). In contrast, all five concentrations of control fish DNA showed amplified bands, with the best resolution when 2 µl of aliquot (ca. 2 ng) were added to the PCR reactions. Given the patterns of strong positive amplification of fish DNA in this experiment, 2 ng of fish DNA aliquots were used as controls to test for PCR inhibition in subsequent assays. Foregut aliquots ranging from 6–2 µl from different individual crabs arrested PCR in all cases (Fig. 2). In the absence of any foregut extract, 28s primers amplified fish DNA strongly and the negative control showed this amplification could not be explained by contamination of the mix (Fig. 2). This inhibition was seen when using bacterial 16s primers as well. When midgut, hindgut and gills from three crabs were extracted together and amplified using the bacterial primers, clear bands around 500 bp were observed (Fig. 3). In contrast, when the same mixtures from the same individuals also contained aliquots from the foregut, no bands were observed. A negative control showed no amplification either (Fig. 3). The inhibitor was effective up to one tenth of its concentration in the aliquots as shown in an experiment using 18s primers (Fig. 4). Further dilutions produced clear amplified product bands in the gels on the same expected positions as those from the positive controls (only fish DNA). No contamination of the mix (negative control) was detected (Fig. 4).

Amplification of control DNA with 18s in the presence of foregut extracts was possible if midgut suspensions were present in the mix, thus suggesting regulation of the inhibitor in the midgut (Fig. 5). The effect of the midgut component (or components) counteracting the activity of the inhibitor was detected even at $10^{-4}$ dilutions of the original midgut aliquot (Fig. 6). For the sample FG3 + MG3($10^{-4}$) there was only a faint band, suggesting that the dilution of the midgut was approaching the limit of activity against the inhibitor.

**Peer**J

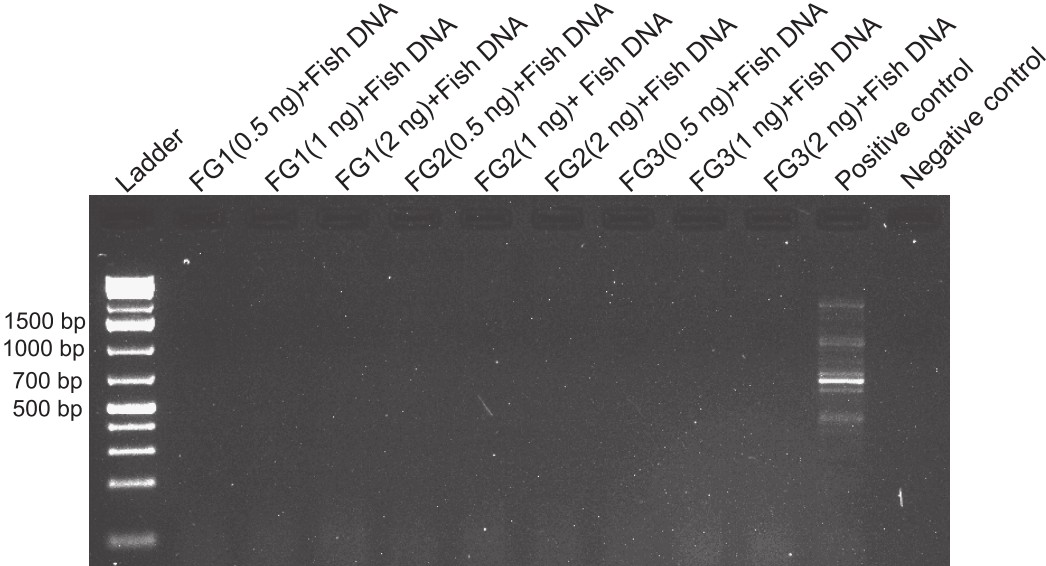

**Figure 2  Effects of foregut extract on fish DNA amplification.** Amplification of fish DNA using 28s primers in the presence or absence of foregut (FG) extracts. Three concentrations of extracted foregut aliquots from three different crabs were added to the PCR reactions (initial undiluted reactions contained 2 ng of crab foregut and 2 ng of fish DNA extracts per reaction). The positive control contained only fish DNA for amplification and the negative control (to assess potential contamination with foreign DNA) contained no crab or fish DNA.

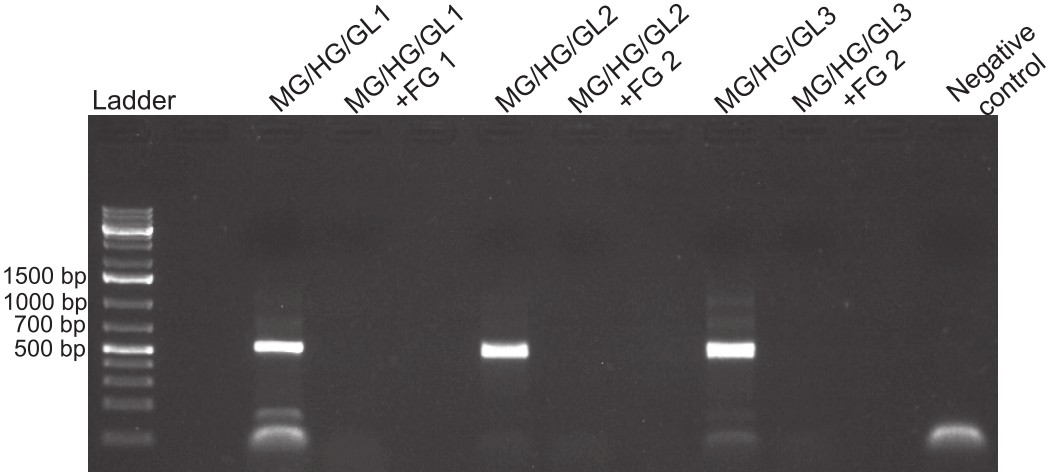

**Figure 3  Foregut inhibition of 16s amplification.** Results of the amplification of bacterial DNA in crab tissues using universal 16s primers (see Methods). Samples from three different crabs were prepared to comprise either mixtures of midgut (MG), hindgut (HG) and gills (GL) together, or these three combined with foregut (FG) extracts from the same respective individual. The negative control assessed potential bacterial contamination and contained no crab or fish DNA. Empty lanes in the gel are not labeled.

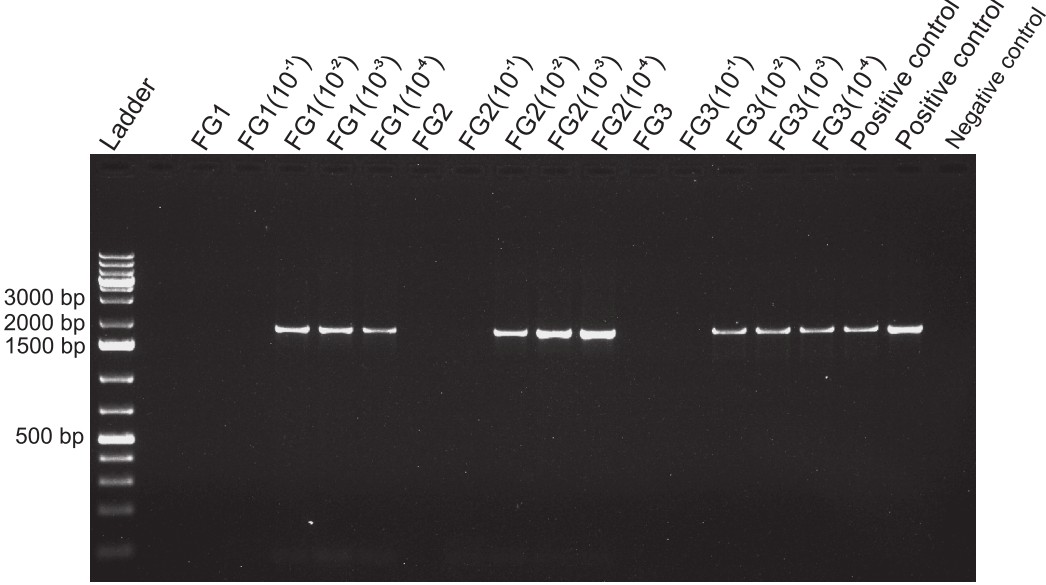

**Figure 4  Activity limits of foregut inhibitor.** Dilution experiment using 18s primers to determine the activity limits of the inhibitor. The initial concentration of each foregut (FG) extract ($N = 3$) was serially diluted ($10^{-1}$ to $10^{-4}$) for a total of five test concentrations per crab foregut. One empty lane (next to the ladder) is not labeled. The two positive controls contained only fish DNA (2 μl) added to the PCR mix and no DNA was added to the negative control to assess potential contamination of the reaction mix.

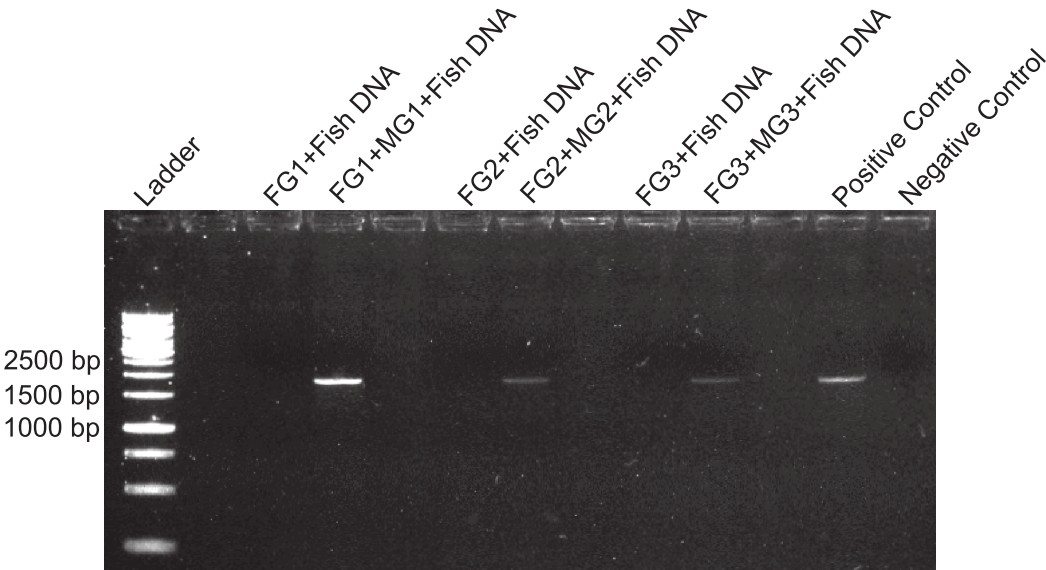

**Figure 5  Midgut degradation of inhibitor.** Amplification of 18s sequences from fish in the presence of foregut (FG) extracts and FG mixed with midgut (MG) suspensions from the same individuals ($N = 3$). Bands of PCR product indicate the neutralization of the foregut inhibitor. Positive and negative controls are as in Fig. 4. Empty lanes are not labeled.

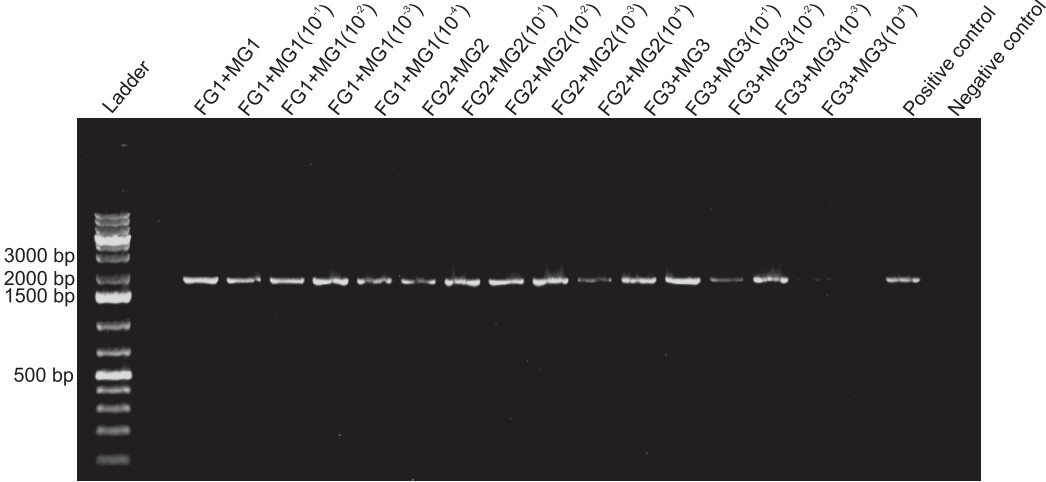

**Figure 6 Dilution of midgut aliquots and foregut inhibition of PCR.** Tests of the activity limits of midgut (MG) suspensions to counteract the foregut (FG) PCR inhibitor ($N = 3$). Fish DNA (3 μl ) was mixed with FG extracts and one of five concentrations of MG extract (baseline concentration to $10^{-4}$) from the same crab before amplifying with 18s primers. Presence of PCR product indicates the neutralization of the inhibitor. Positive and negative controls are as in Fig. 4. Empty lanes are not labeled. Though not completely clear from the picture, a very faint band, denoting amplification, was observed for FG3 + MG3($10^{-4}$).

For these two experiments, both positive and negative controls showed the expected patterns (Figs. 5 and 6).

The inhibitor was heat stable as shown by an experiment in which foregut extracts from three individuals crabs were heated for an hour at 99 °C before adding them to the PCR reactions. The addition of these boiled extracts to fish DNA blocked amplification with 18s, while fish DNA without any foregut extract added, amplified at the expected band size (Fig. 7). This suggests that the inhibitor is not a protein with a complex structure that would undergo denaturation. The inhibitor also did not react as expected from a melanin-like compound (*Dörrie et al., 2006*)). When extracts from all three main sections of the crab gut, gills, crab muscle and fish muscle were spectrophotometrically evaluated, no significant differences in absorbance were observed (Fig. 8). In fact, despite sometimes large variance, absorbance for most samples ranged from 0–0.05 and was not statistically different than zero ($P \geq 0.219$, Mann–Whitney U tests). One-way ANOVA showed no significant difference between any of the six treatment groups in terms of potential melanin content ($P = 0.284$; Fig. 7).

## DISCUSSION

A compartmentalized PCR inhibitor was present in the foregut but not other parts of the porcelain crab *Petrolisthes rufescens*. This was clear from the absence of any amplified DNA bands at the expected sizes using 28s, 16s or 18s primers when foregut extracts were present. The inhibition of amplification of both nuclear (28s and 18s) and microbial genes (bacterial 16s) suggests that the activity of the inhibitor is general and potentially effective against both eukaryotes and prokaryotes. Excess or critically low amounts of DNA in the

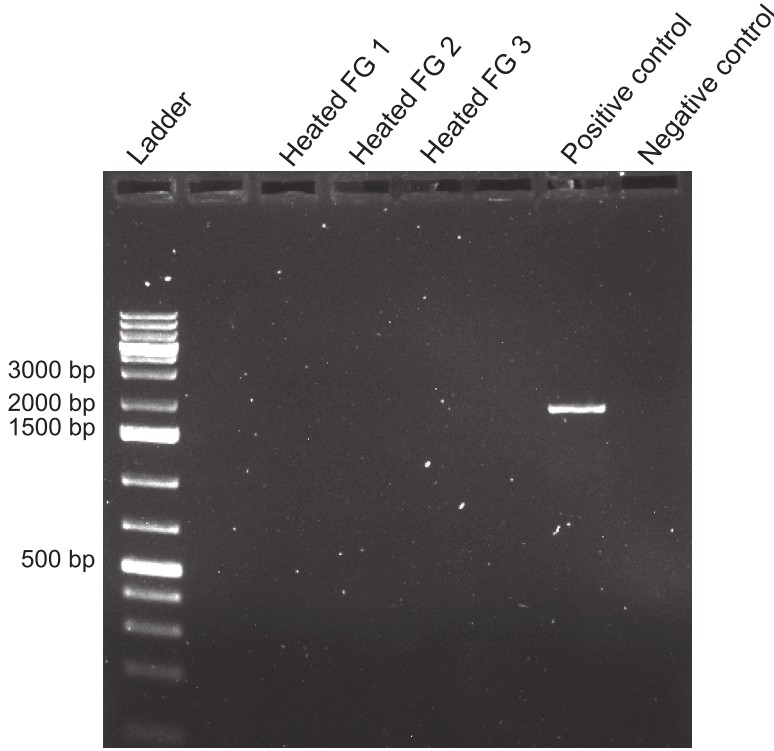

**Figure 7  Effect of boiling on foreguts.** Effects of heating foregut (FG) extracts (99 °C, 1 h) to assess the stability of the inhibitor. Three different foreguts were boiled and added to fish DNA before amplifying with 18s primers. Empty lanes are not labeled. Labeled lanes without bands indicate inhibition of PCR. Positive and negative controls are as in Fig. 4.

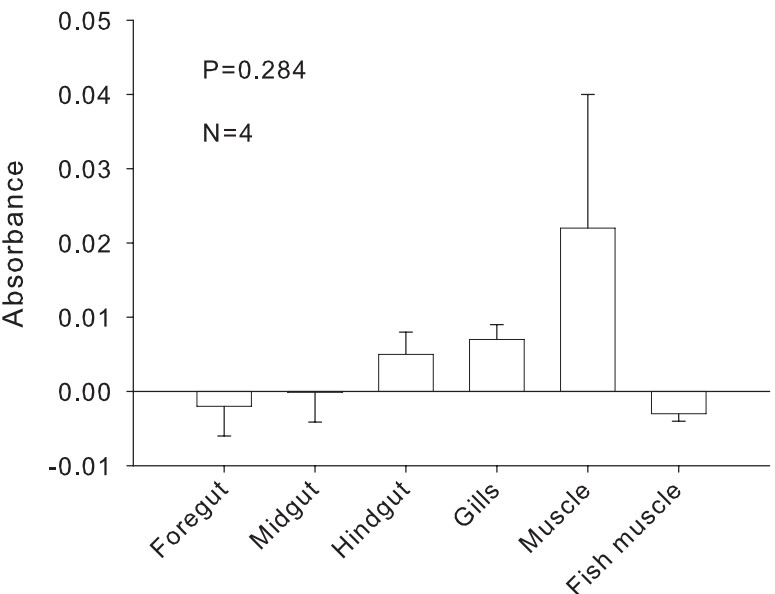

**Figure 8  Relative melanin content of samples.** Absorbance of samples at 320 nm to assess melanin content ($N = 4$). Bars represent means $+$ 1SE. Statistical analysis was performed with one-way ANOVA.

samples did not explain inhibition of PCR (*Wilson, 1997*; *McBeath et al., 2006*; *Pan et al., 2008*; *Alaeddini, 2012*). Firstly, NanoDrop measurements showed that the amounts of DNA in extracted foregut samples compared to those extracted from other crab tissues and from fish (Table 1). Second, all dilutions of all samples of foreguts in the first experiment (which evidently contained DNA) failed to amplify using 28s primers, whereas the five dilutions of fish DNA amplified well (Fig. 1). Dilution has been successful in reducing or eliminating the effects of inhibitors in PCR either by reducing the amount of the inhibitor or of the DNA template (*Wiedbrauk, Werner & Drevon, 1995*; *Mättö et al., 1998*; *McBeath et al., 2006*; *Pan et al., 2008*; *Alaeddini, 2012*). The fact that DNA was extracted from foreguts at comparable concentrations to those of samples which amplified suggests a highly active or concentrated inhibitor in *P. rufescens*. That the inhibitor was found only in the foregut (or at least found in effective concentrations to arrest PCR) was confirmed by amplifying bacterial 16s rRNA genes from midgut and hindgut, along with gills. Clear amplification only happened when foreguts were absent from the mixture (Fig. 3).

When dilutions were used to assess the lower limits of activity of the inhibitor, inhibition was still observed at one order of magnitude below the initial extracted amount (Fig. 4). Although we do not show these data, similar results were observed by decreasing progressively in half the amount of aliquot used in the reactions and amplifying with 28s primers (Fig. S2). In that experiment, when fish 28s DNA was amplified in the presence of crab foregut extracts all the three dilutions of the foreguts inhibited amplification, whereas the control, lacking foregut extracts, amplified. The mass of extracted foreguts for the dilution experiment above was known (26–42 mg). However, the foregut of crustaceans is lined with a cuticle that is largely comprised of chitin (*Brunet, Arnaud & Mazza, 1994*; *McGaw & Curtis, 2013*; *Watling, 2013*). Thus, the amount of actual tissue that could secrete and/or store the inhibitory molecule was unknown. Experiments using real time-QPCR (*Schneider, Enkerli & Widmer, 2009*) or internal positive controls for PCR (*Hartman, Coyne & Norwood, 2005*) could provide better resolution for the activity of the inhibitor and its effective concentration. The conclusive determination of the precise inhibitory molecule and its concentration, however, will require chemical microtechniques that will allow elucidation of components in very small tissue samples as to reduce collection impacts on porcelain crab populations.

While the identity of the molecule responsible for the observed results remains unknown, the inhibitor did not appear to be a protein or peptide. Neither increasing proteinase K added to the samples during extraction (see Methods), or heating foregut samples at 99 °C for 1 h before amplifying by PCR, reduced the inhibitory activity (Fig. 7). Medical studies aimed at detecting pathogens in saliva and ocular fluids have found similar results (*Ochert et al., 1994*; *Wiedbrauk, Werner & Drevon, 1995*; *Mättö et al., 1998*). *Ochert et al. (1994)* concluded that PCR inhibitors in their saliva samples were likely carbohydrates, based on the addition of proteinase K, extraction in phenol-chloroform, treatment with ion-exchange resins, and boiling of the samples. Our results similarly suggest a carbohydrate inhibitor, although it is unknown if small glycopeptides or other mixed-origin molecules can survive proteinase K digestion. It should also be noted that,

in one case, PCR inhibition by a protein (immunoglobulin G) has been shown to increase with temperature (*Abu Al-Soud, Jönsson & Rådström, 2000*).

We can, however, dismiss the occurrence of another group of known PCR inhibitors: the melanins (*Yoshii et al., 1993*; *Eckhart et al., 2000*; *Opel, Chung & McCord, 2009*). These compounds are abundantly found in invertebrates, including crustaceans (*Bandaranayake, 2006*; *Dubey & Roulin, 2014*), where they are associated with processes of wound repair and immune response (*Söderhäll, 1982*; *Sritunyalucksana & Söderhäll, 2000*; *Vázquez et al., 2009*; *Dubey & Roulin, 2014*). The mechanisms of PCR inhibition by melanins have been explicitly studied. These include binding directly to polymerases (*Eckhart et al., 2000*) or reducing the amount of template available for amplification by binding to specific DNA sequences (*Opel, Chung & McCord, 2009*; *Geng et al., 2010*); longer amplicons may be more sensitive to the effects of melanin than short ones (*Opel, Chung & McCord, 2009*). Using a spectrophotometric technique (*Dörrie et al., 2006*) we compared samples of *P. rufescens* foregut, midgut, hindgut, gills, and muscle, and fish muscle (Fig. 8). No significant differences in absorbance were found among tissues, suggesting that the inhibitor was not a melanin-like compound localized in the foregut. While we did not attempt to quantify the potential amounts of melanin-like compounds in the tissues (using a purified standard), absorbances were close to zero and the most variable signal was found in the crab muscle, from which we have readily amplified bacterial sequences in other experiments (Fig. S3).

A functional molecule is expected to be regulated by the organism containing it. Preliminary observations showed that microbial sequences could be recovered from other parts of the digestive tract of *P. rufescens*. Therefore, we hypothesized that regulation of the inhibitor would occur in the midgut. This was confirmed when control fish DNA was amplified successfully in the presence of foregut extracts *only* if they were mixed with midgut suspensions. Two lines of evidence suggest that the regulatory molecule is an enzyme or at least a protein in the midgut. First, midgut suspensions did not counteract the inhibitor when proteinase K was added to the PCR mix (Fig. 3). In contrast, when proteinase K was absent, amplification of DNA was possible even at $10^{-4}$ dilutions of midgut suspensions (Fig. 6). Second, the midgut contains secretions from the hepatopancreas, a multiple-function digestive and absorptive organ that produces a variety of digestive enzymes (*Brunet, Arnaud & Mazza, 1994*; *McGaw & Curtis, 2013*; *Watling, 2013*). The secretory ability of the hindgut is mainly restricted to producing mucus to facilitate the expulsion of feces. Some ion uptake may also occur, but a role in enzymatic degradation of the inhibitor is unlikely.

The location of the inhibitor suggests a potential role in immunity and pathogen control for *P. rufescens*. It may also serve as a bottleneck for the control of gut flora by selecting for specific microbes to colonize the gut. The foregut is among the first internal compartments in contact with ingested water and particles from the outside environment. Hence, it is a prime route through which pathogens can gain entrance (*Small & Pagenkopp, 2011*). Being a filter feeder, *P. rufescens* may have some ability to control for size of ingested particles, but it is unlikely to control the nature of microbes attached to them. The inhibitor could serve as a broad action barrier to reduce pathogen invasion. DNA polymerization

is fundamental for cell proliferation and some natural antibiotic compounds are known to provide immunity to plants, invertebrates and vertebrates (*Zasloff, 2002*; *Brogden et al., 2003*; *Tincu & Taylor, 2004*; *Otero-González et al., 2010*; *Brandenburg et al., 2012*) by inhibiting DNA replication, among other mechanisms (*Boman, Agerberth & Boman, 1993*; *Brogden, 2005*; *Brandenburg et al., 2012*). Most of these compounds are small peptides, which could potentially survive extraction methods such as ours. Although the majority of these have been isolated from hemolyph and hemocytes, some are produced in the buccal cavity or digestive tract of the organisms, including humans (*Boman, Agerberth & Boman, 1993*; *Gorr, 2009*; *Brandenburg et al., 2012*).

It is unclear how the crab avoids the negative effects of the inhibitor within its foregut tissues. One possibility is that the cuticle lining the foregut (*Brunet, Arnaud & Mazza, 1994*; *McGaw & Curtis, 2013*; *Watling, 2013*), serves as a barrier preventing the widespread penetration of the inhibitor into the crab cells. Exclusion at the level of the cell membrane could also explain this. Another relevant question is how gut microbes in the midgut and hindgut circumvent the inhibitor. Clearly, chemical isolation of the inhibitor (and its counteracting molecule in the midgut), and definitive proof of the foregut inhibitor effects *in vivo* are needed before its natural role can be conclusively determined.

Our findings have implications for the design of studies using molecular tools to detect small organisms in environmental samples, and for those assessing gut microbial diversity in crustaceans and other animals. The potential to obtain false negatives from molecular probes is higher for small organisms in which tissues or organs cannot be clearly separated and are, therefore, processed as whole animals. Inhibitors can obscure the detection of larval crustaceans and gastropods (*Vadopalas et al., 2006*; *Jensen et al., 2012*), as well as parasitic copepods (*McBeath et al., 2006*), from plankton samples. Other inhibitors hamper PCR on the resting eggs of fairy shrimp (*Moorad, Mayer & Simovich, 1997*), the eyes of bees (*Boncristiani et al., 2011*) and when detecting pathogens in shrimp tissues (*Wang, Hong & Lotz, 1996*). While an array of inhibitors has been studied from biomedical studies, food microbiology, and forensics (*Wilson, 1997*; *Maurer, 2011*; *Alaeddini, 2012*; *Schrader et al., 2012*), we suspect such information from other fields is often relegated to unpublished observations about experiments that did not work among scientist notes. Clearly, there is a practical relevance to the study of inhibitors in order to create protocols for optimizing PCR (*Wilson, 1997*; *Rådström et al., 2004*; *Alaeddini, 2012*), but such inhibitors may also provide leads into drug discovery. Polymerase inhibitors have been, and continue to be, studied as potential agents against microbial pathogens and cancer, for which arresting cell proliferation is a fundamental step in treatment (*Liu-Young & Kozal, 2008*; *Javle & Curtin, 2011*; *Gane et al., 2013*). To date, however, the ecological and evolutionary relevance of these inhibitors has been largely overlooked.

## ACKNOWLEDGEMENTS

This research was produced in partial fulfillment of an undergraduate senior thesis from the Biology Department of The American University in Cairo by Edwin Cruz-Rivera. We gratefully acknowledge Mohie El-Din Sherif and Tamer Hafez for their help

collecting specimens and Hamza El-Dorry for suggestions and the use of equipment. We also thank Arthur Bos for providing the fish for DNA extraction. Comments by Pietro Gatti-Lafranconi, Michael Sweet and Christina Eichstaedt greatly improved this manuscript.

### Funding

Funding was provided by a Faculty Research Grant from the American University in Cairo to Edwin Cruz-Rivera, and by the King Abdullah University for Science and Technology Global Collaborative Partners (GCR) program, which supported Ari Ferreira and Amged Ouf. The funders had no role in study design, data collection and analysis, decision to publish, or preparation of the manuscript.

### Grant Disclosures

The following grant information was disclosed by the authors:
The American University.
King Abdullah University for Science and Technology Global Collaborative Partners (GCR).

### Competing Interests

The authors declare there are no competing interests.

### Author Contributions

- Mahmoud A. El-Maklizi and Amged Ouf conceived and designed the experiments, performed the experiments, wrote the paper, prepared figures and/or tables, reviewed drafts of the paper.
- Ari Ferreira conceived and designed the experiments, analyzed the data, contributed reagents/materials/analysis tools, wrote the paper, reviewed drafts of the paper.
- Shahyn Hedar performed the experiments, reviewed drafts of the paper, collected specimens and maintained lab animals.
- Edwin Cruz-Rivera conceived and designed the experiments, analyzed the data, contributed reagents/materials/analysis tools, wrote the paper, prepared figures and/or tables, reviewed drafts of the paper.

### Supplemental Information

Supplemental information for this article can be found online at http://dx.doi.org/10.7717/peerj.689#supplemental-information.

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
