# Peer review of "A localized PCR inhibitor in a porcelain crab suggests a protective role"

_PeerJ, doi:10.7717/peerj.689_

## Round 0.1 · original submission · Minor Revisions

Both referees are supportive of this submission and I find myself of the same opinion. The manuscript cannot, however, be accepted for publication in its current form, primarily because of concerns regarding the methodology followed.
The authors will find that both reviewers recommend a number of actions to improve the clarity and reproducibility of their experimental approach. In particular, negative controls and accurate DNA concentration values have to be provided to support the authors' claims. These have been identified as the main critical points by both reviewers, and by myself.
I encourage the authors to provide this missing information (that should require a minimal amount, if any, of additional experiments) and edit the remaining of the manuscript to accommodate the reviewers' comments. I believe the paper would benefit from these minor changes, and a revised version be suitable for publication.

·

Basic reporting

This study entitled ‘a localised PCR inhibitor in a porcelain crab suggests a protective role’ is an interesting topic largely unreported in the scientific literature. It utilises simple PCR techniques to highlight as yet unknown inhibitors is specific areas of crab gut in comparison to fish dna. Although i recommend this study for publication, there are several issues I would like to see addressed, particularly with focus on the methodology utilised. It may be the case that further lab work is necessary to fulfil some of these, however it would mean that the conclusions drawn are that much stronger. Below I outline these specific suggestions but understand that some may not be achievable as this was largely an undergraduate project and likely further research cannot be conducted.
Intro
Page 3 Line 43 sentence ending in block PCR needs a reference
Page 3 Line 52 reference at end of sentence. The next sentence on this line shouldn't start with because – not true English.

Discussion
I personally don’t like referencing figures within the discussion section. The way the discussion is currently structured feels more similar to your results section. Please have a careful look through this section and make sure you relate your findings more to previous studies published for example P10 L 25 sentence ending prokaryotes. Are there other studies which have shown this? Where is your reasoning behind this statement?
P11 L37 insert rRNA after 16S
P11 L 41 again another pet peeve of mine is the reporting of unpublished data, why not include this in the publication, if not vital then either omit or add as supplementary materials. The problem here is that reviewer and then the readers cannot look at these results and assess the validation of them. I have the same issue with referencing unpublished studies, which you do in the case of the bacterial communities associated with the crabs. I would either omit this of wait till the other study is published depending on the stage of this other publication P13 L79 for example.
P12 L60 Anyway you could assess extraction via phenol-chloroform. These results would add validation to your claims at this point and relate specifically to the study you mention here. I believe, although increasing the amount of work you need to do, this extra lab work will assist in the quality of this study significantly.
P12 L 71 sentence starting with Longer, this seems a little out of place, can you link these sentences together a bit more to improve connectivity.

Experimental design

Methods
This is where I would like to see considerable clarification on numerous issues. Throughout the text the authors make reference to using a specific amount of DNA in µl. this could be a major flaw in this study as you should really utilise ‘ng’ of the DNA to standardise not µl. Specifically as the authors then go onto the state that there is significant variation in DNA quality and quantity between replicates of the same sample type and between the different samples themselves. An alternate interpretation of this data could simply be down to variations in starting DNA. Although I don’t believe this to be the case it needs to be addressed.
Specifically see comments below;
Page 5 Line 07 ‘ a fixed amount’ how much?
Page 6 Lines 20-23 you utilised a low number of replicates throughout, n = 3 for each treatment. Any reason for this? Can you repeat the study to strengthen your arguments?
A negative control of PCR mix? What PCR mix was utilised and a negative control should have also utilised the elucidation agent ie. A buffer or sterile water when you extracted your DNA.
P6 L 30 why did you not use the same concentrations you used for the crabs for the fish controls?
P8 L 61 reference at the end of the sentence to support your statement
Check throughout for changes in font as there appears to be size error in latter end of methods
P8 L74 and 45 reference at the end of these two sentences
Results
P8 L 80 the variation you find in your extracted DNA highlight the issues relating to the methods utilised and the importance in standardising your work using ng over ul
P10 L 16 please put statistics here in text to back up ‘not statistically different’

Validity of the findings

the findings are important however there are some major issues with the methods utilised see above section

Despite these slight issues I have with methodology, I would recommend publication of this study, however a more detailed study addressing the issues raised above may be more valid in the future.

·

Basic reporting

The article reports findings of an PCR inhibitor located in the crab foregut. It thoroughly shows its localised presence and its removal by midgut extracts.
The background is well described and discussed.

There are a few minor comments:
1. Please use consistent spelling of the ribosomal RNA genes, the "s" should either be capitalised throughout the manuscript or not.
2. Line 52
Instead of "serendipitous" "incidental" might better represent the meaning of the sentence.
3. Line 74 "The" should not be capitalised
4. e.g. Line 140
What is meant by "full strength" extracts? Undiluted?
5. e.g. Line 160
font size should be consistent. A few sentences and references have a smaller font size.
5. Line 300/301
a "to" is missing (known to)

Figure 1:
The visibility of bands should be increased if possible. The bands are in particular faint in comparison to the other gel pictures.

General comment:
A table in the method section could clarify the different approaches.

Experimental design

1. Line 83:
It is unclear to the reader why some body parts were extracted from 6 or from 3 crabs. This contradicts the supplementary table showing the extraction of all body parts from 6 crabs. If body parts from 6 crabs were extracted, how were the respective extracts chosen for experiments in which only 3 extracts are presented? By DNA concentration?
2. e.g. Line 100
when "extract" is used, does it always refer to DNA extract? If yes, it would be useful to add this information.
3. Line 150
Why was an incubation temperature of 42°C chosen? Does that realistically represent the gut temperature of the crab in its natural habitat?
Figure 2:
The 28S primers seem very unspecific. A better designed primer might yield more specific bands.

Line 256/286:
The use of Proteinase K is not well described in the methods. Please provide information to which set of PCRs it was added. The mentioning of Proteinase K is also missing in the legend of Figure 3, it only comes up in the discussion, where the reader is referred to the methods and to Figure 3.
General comment:
Please provide PCR conditions to allow reproducibility of experiments.

Validity of the findings

Was there a negative extraction performed? Considering the very low amounts of DNA extracted from some body parts it would be very interesting what "concentration" a negative extraction might yield.
E.g. Line 178 states a minimum of DNA of 0.003 ng/µl was extracted, which might mean DNA extraction of this particular sample was unsuccessful.

Additional comments

The authors provide a rigorous assessment of PCR inhibitory effects of the foregut extract. With additional clarifications this manuscript will provide an interesting addition to the literature.

---

## Round 0.2 · Minor Revisions

The Authors edited the manuscript to accommodate reviewers' comments, and this has now been greatly improved.
The level of detail used to describe experimental procedures is however still a cause of concern for one reviewer, and this observation finds me in agreement.

In summary, the two key aspects that need to be addressed are:

1. Reporting (and reproducibility)
There is no reason why the amount of DNA used in each reaction should not be expressed in terms of ng/reaction. These values will either be small (e.g. Fig. 1) or will be expressed as dilutions of a reference solution (e.g. Fig. 4). The manuscript represents a useful resource for scientists trying to optimise extraction/amplification procedures on similar samples, but its utility is compromised if tested DNA concentrations cannot be reproduced. DNA concentrations should then be converted to ng throughout the manuscript, and added where missing (e.g. Figure S3). Similarly, the concentration of the 'Fish DNA' control sample should be added to table 1.

2. Sensitivity and false positives
The extraction procedure leaves the authors with very low amounts of DNA, so low they could result from a minor deviations of background absorbance in NanoDrop reads (also, the number of significant figures reported in Table 1 seems to be way beyond the sensitivity limits of the instrument). Although the authors' results are consistent with DNA being present in all samples (but foregut extracts, by definition) despite the low concentrations detected, how very minor variations in the amount of template DNA (roughly between 1 and ~5 ng/reaction, i.e. in the range of technical variability) can have such dramatic effects on PCR efficiency is more difficult to explain. Variability between DNA extracts (often of the same order of magnitude than the average) and their purity could account for the observed changes, particularly considering the low absolute values. I thus echo the request for a clear acknowledgement of such problems in the manuscript.

In conclusion, this study suggests a better understanding of the chemical composition of crab DNA extract could provide information on elaborate defence mechanisms. In addition, it also provides a protocol for DNA amplification from relatively uncharacterised sources. However, confidence in the methods used has to reach a higher standard for both of these goals to be achieved.

·

Basic reporting

Generally ok, still some issues regarding methodology, see below.

Experimental design

2 ng of DNA is quite low in my opinion, furthermore the corrections slightly worried me i.e. that 2 ul equates to 2 ng across the board, as the authors state there was variation in the amount of DNA extracted from the samples '0.003(in one midgut) ng/μl to a
maximum of 9.53ng/μl'

what does Line 95 mean 'We annotate our results from gels as volumes for the purpose of consistency because many experiments entailed serial dilutions.' can you reference another paper which has done this to back up your reasoning behind this technique.

Line 103 'too high or too low concentrations of DNA' ? at 2ng this is unlikely to be too high and how can it be too low if its all the same conc used throughout.

Line 106 for the same reasoning as in the first review using μl of DNA is not very useful to others trying to replicate your results 'Amounts of 6μl, 4μl, and 2μl of DNA from initial concentrations were used'.

Line 152 how much/ 'These were compared to samples from the same crabs in which foreguts and midguts were mixed in equal amounts before adding the fish DNA'.

Validity of the findings

with the issues stated above I remain slightly reserved over the findings from this study.

Additional comments

Although I am slightly worried about the methodology, I believe it will assist others in their work and therefore should be published. I would however like acknowledgement of the potential problems of this manuscript in the abstract, results and discussion so readers are made aware that there are some sampling issues and therefore results should be taken with caution.

·

Basic reporting

The article is well written.

Experimental design

The design is clearer now.

Validity of the findings

no comments

Additional comments

I support the revised manuscript to be published.

---

## Round 0.3 · accepted · Accept

The authors's rebuttal to the latest reviewers comments clarified the methodological aspects of the manuscript further. The 'cautionary tale' included in this paper is now well documented, and hence provides a useful resource for future studies.